# The Anomalous Behavior of Thermodynamic Parameters in the Three Widom Deltas of Carbon Dioxide-Ethanol Mixture

**DOI:** 10.3390/ijms22189813

**Published:** 2021-09-10

**Authors:** Evgenii Igorevich Mareev, Alexander Petrovich Sviridov, Vyacheslav Mihailovich Gordienko

**Affiliations:** 1Federal Scientific Research Centre “Crystallography and Photonics”, Institute of Photon Technologies, Russian Academy of Sciences, Pionerskaya St. 2, Troitsk, 108840 Moscow, Russia; sviridoa@gmail.com (A.P.S.); v_m_gord@mail.ru (V.M.G.); 2Faculty of Physics, M. V. Lomonosov Moscow State University, Leninskie Gory Bld. 1/2, 119991 Moscow, Russia

**Keywords:** supercritical fluids, mixture, molecular dynamics, clusters, Widom delta

## Abstract

Using molecular dynamics, we demonstrated that in the mixture of carbon dioxide and ethanol (25% molar fraction) there are three pronounced regions on the p-T diagram characterized by not only high-density fluctuations but also anomalous behavior of thermodynamic parameters. The regions are interpreted as Widom deltas. The regions were identified as a result of analyzing the dependences of density, density fluctuations, isobaric thermal conductivity, and clustering of a mixture of carbon dioxide and ethanol in a wide range of pressures and temperatures. Two of the regions correspond to the Widom delta for pure supercritical carbon dioxide and ethanol, while the third region is in the immediate vicinity of the critical point of the binary mixture. The origin of these Widom deltas is a result of the large mixed linear clusters formation.

## 1. Introduction

In recent years, supercritical fluids (SCF) have become the object of close attention due to their wide applications in technological processes of purification, separation, and extraction of various substances, in such areas as biotechnology, food industry [1], propulsion systems [2], liquid-propellant rocket engines, high-pressure gas, nanoparticle and cluster production [3,4], and supercritical chromatography [5]. Significant progress has been achieved in understanding supercritical fluids’ physical and chemical properties [6,7], primarily due to improved experimental techniques [8,9]. However, several questions regarding the physical micro- and macroscopic behavior of supercritical fluids remain open. In the phase diagram, the line separating the liquid from the gas phase ends in the critical point. Previously, it was believed that the supercritical state is homogeneous. Recent experiments identified areas with different liquid or gaseous properties even under supercritical conditions [10]. Indeed, above the critical point in the p-T diagram, a change in the structure of the substance is also observed: from weakly interacting molecules and their clusters (gas-like SCF) to “superclusters” with “interspersed” regions of almost empty space (liquid-like SCF) [11,12,13,14]. The change in the SCF structure from liquid-like to gas-like occurs during passing through the coexistence frontier, separating noticeable changes in the state of the medium. This so-called Widom delta (due to the delta-like shape on the p-T diagram) was first identified experimentally by K. Nishikawa and I. Tanaka [10]. They demonstrated the difference of the supercritical state from the classical representation as having homogeneous properties. The Widom region is usually associated with the thermodynamic transition of a single-component system. T. Sciortino et al. [15] introduced the Widom region as a set of states with the maximum correlation length [7,16]. Still, often this area is approximated as the locus of points on the p-T diagram, where the maximum thermodynamic parameters are reached since they can be more easily estimated [17,18,19,20]).

From the known results [21], it follows that impurities shift the location of the Widom delta in supercritical CO_2_ [22] and H_2_O fluids [23]. Such investigations were carried out for several individual substances, such as CO_2_ [24,25,26,27,28], O_2_ [29], Ar [7,30], and H_2_O [31], as well as mixtures of noble gases. Crucially, that the addition of even a tiny amount of another substance with different critical pressure and temperature can significantly shift the critical parameters of this system (up to hundreds of degrees and bars), because in a first approximation, the critical temperature and the critical pressure of the resulting mixture can be estimated as partial values of the critical parameters of each component. In addition, the presence of two components in the system significantly complicates the interaction between molecules and particles. This interaction ensures the appearance of mixed clusters and anomalous behavior of thermodynamic (TD) parameters. In our previous paper [32], we revealed that, in the CO_2_–ethanol mixture, large mixed clusters are formed, which leads to the increase of nonlinear refractive index (n_2_). The effect of anomalous n_2_ increase could find applications in nonlinear optics. However, there are still a lot of open questions, primarily regarding the impact of clusters on the macroscopic properties and the whence of the Widom delta. Nowadays, there is a gap in describing the thermodynamics of transition states in multicomponent systems in addition to identification of various phases. The range of existence of the Widom delta for binary mixtures is of great technological importance. In particular the CO_2_–ethanol mixture is applied in the process of cannabis extraction [33], fish oil extraction [34], caffeine extraction [35], etc. The increase of solubility that is achieved in the Widom delta (due to clustering) can significantly increase the efficiency of these processes [36]. It is also worth mentioning that another substance can be initially dissolved in ethanol, and due to the effective diffusion of SCF carbon dioxide, the substance then spreads throughout the system [37]. Therefore, it is important to predict the range of optimal pressures and temperatures for extraction, ablation, nanoparticle production, etc.

In this work, we have expanded the well-proven molecular dynamics methods (MD) [38] to study the behavior of the thermodynamic parameters of a carbon dioxide and ethanol mixture (density, enthalpy, specific isobaric heat capacity), as well as the microstructure of the medium (parameters of clustering of the medium) in the Widom delta vicinity.

## 2. Results and Discussion

Before discussing numerical simulation results, it is necessary to recall that a mixture of carbon dioxide with ethanol in a molar ratio of 3:1 under room conditions is a two-phase system [39,40]. In such a mixture, liquid ethanol with a small amount of dissolved carbon dioxide molecules in it is surrounded by gaseous carbon dioxide. MD calculations show that at temperatures below critical for carbon dioxide (for example, 275 K) and atmospheric pressure, carbon dioxide dissolves well in ethanol (the calculated enthalpy of mixing is of the order of −4 J/mol, which is comparable with the tabular −3.7 J/mol [41], the free energy of mixing is about −1.5 J/mol). Figure 1a also shows a uniform distribution of ethanol and carbon dioxide molecules. A significant amount of carbon dioxide molecules (about 50%) is dissolved in ethanol, forming a quasi-homogeneous mixture. Another part of the molecules is in the gas phase outside the formed supercluster. Below the critical pressure and temperature, with an increase in temperature at constant atmospheric pressure, the isolated structure (the dissolved in the ethanol CO_2_) begins to pass into a gaseous state. With an increase in pressure up to about 40 bar (at room temperature), the mixture turns into a liquid state [42]. However, the transition to the supercritical state of matter significantly complicates the microstructure of the mixture. In the case of a one-component system (pure CO_2_), the single Widom delta is achieved (see Section 3 3). In this region, a maximum of density fluctuations is reached, and abnormal behavior of thermodynamic (TD) is observed. Moreover, in this region, the liquid-like SCF transforms into a gas-like SCF.

As we showed in [32], supercluster decay is observed in the Widom delta, which is characteristic of a liquid state of matter and a liquid-like SCF. Therefore, at the first stage, the MD method was used to study the dependence of the medium density (ρ) and the maximum clusters size (S_max_) on pressure and temperature. In this work, the Widom region was identified as a region with maximum density fluctuations, accompanied by the disintegration of the supercluster.

For pure carbon dioxide, the Widom region can be identified by determining the maximum dependence of the partial density derivative on pressure for different temperatures. As shown in Figure 2, on the heat map of the density derivative on pressure δρ/δp (p, T), there is a region where the maxima of this derivative are observed. This graph was constructed based on the data [43], with the following numerical differentiation. The physical processes associated with the Widom delta (an extremum of thermodynamic parameters, maximum density fluctuations, and cluster formation) are most clearly manifested near the critical point. At a longer distance, the experimental conditions on the p-T (pressure–temperature) diagram from the critical point, the peak of density derivative on pressure becomes wider (Figure 2) and its amplitude is decreased. The maximal amplitude is achieved for a pressure of 79 bar and a temperature of 308 K.

Figure 3a shows heat maps of the mixture density dependence on pressure and temperature. The local density was calculated using Radical Voronoi tessellation, which was previously successfully applied for characterization of the gas-like and liquid-like SCF [44]. The tessellation was performed with Voro++ library [45]. Several pronounced areas can be distinguished in this figure. The first region corresponds to a two-phase medium; it is in the region of relatively low pressures (less than 40 bar) and temperatures (less than 400 K), see Figure 1a. At higher pressures, most of the mixture is in the liquid phase. However, in the vicinity of 310 K at pressures above 80 bar, a decrease in the mixture density is observed (a solid curve marks the area in Figure 3a). Furthermore, in the vicinity of 410 K, a local density maximum is observed, and in the region of 420–460 K, a rapid (2.5 times) decrease in density is observed at pressures above 60 bars. The Widom delta of carbon dioxide is observed for temperatures about 310 K and a pressure of about 80 bar. Similar behavior is also observed for other thermodynamic (TD) parameters.

The maximum of calculated cluster size (S_max_) equals to 9260 molecules. It is achieved in the region of maximum density. In this region, the structure of the mixture is close to that of a liquid (i.e., infinite supercluster [46]). As a result, most molecules from one massive cluster called a supercluster. The Widom delta corresponding to pure carbon dioxide is visible in Figure 4. It shows a decrease in the maximum size of the supercluster by about 1/3 (from 9260 to 6030 molecules). However, this region is less pronounced than the Widom region of the mixture (in the vicinity of 420 K). In the area with the maximum cluster size, carbon dioxide and ethanol form one dense structure with an almost uniform distribution of ethanol molecules over the system’s volume (see Figure 1c), forming a homogeneous medium. In the Widom delta of carbon dioxide, CO_2_ molecules form separate clusters (see Figure 1) containing only CO_2_ molecules. Due to the intermolecular interaction between the mixture components (in high pressure, more than 80 bar), some carbon dioxide molecules remain in the supercluster, and the clustering process does not develop as efficiently as in pure CO_2_. It is essential to note the area shown by the dashed curve in Figure 4. In the microstructure of the system corresponding to this region, we can find large linear mixed clusters (see Figure 1d). As we revealed in [28], the formation of linear clusters leads to the anomalous behavior of the optical and nonlinear optical properties of the medium in the Widom delta. The Widom delta corresponding to ethanol is less pronounced than the Widom delta corresponding to carbon dioxide or the mixture due to weaker interaction between ethanol molecules. In addition, only 25% of ethanol is in the mixture, and secondly, most of the interactions occur between different molecules.

Since the Widom delta was initially defined as the region in which fluctuations reach a maximum [47], density fluctuations were retrieved using molecular dynamics. The procedure of fluctuations calculation was performed based on the atomic volume occupied by each molecule (which gives the values of the local density). This value was averaged; the variance was calculated and subsequently normalized to the average density value. A three-dimensional heat map of density fluctuations is shown in Figure 5. It shows that fluctuations have the lowest values in the liquid phase (pressure above 80 bars, temperature below 320 K). There is also a region of a gas-like supercritical state, at a pressure above 60 bars and a temperature higher than 480 K. The most increased fluctuations (more than twice the average values) are achieved in the Widom delta for the mixture (pressure over 60 bars, temperature over 420 K) and in transition from two-phase system to single-phase system (curve starting at 20 bar and 260 K and ending at 100 bar and 325 K). The Widom delta of individual components of the mixture is less pronounced: in the region of 340 K and 80 bars for carbon dioxide, and 560 K and 100 bars for ethanol.

It should be noted that fluctuations in the gas phase are more intense than in the liquid one. The rise of fluctuations is due to the relatively weak interaction between individual molecules, which leads to the appearance of local inhomogeneities in the system. Density fluctuations are primarily associated with the microstructure of the medium when, due to the formation of mixed clusters, regions with both high and low density are formed in the medium (see Figure 1d). If we consider the Widom delta as a region separating the liquid-like SCF from the gas-like SCF, then it is here that an intermediate case is realized, for which the coexistence of the liquid-like SCF (cluster-regions of increased density) with the gas-like SCF (rarefaction region) is characteristic. Growth of density fluctuations (in comparison with pure media) will lead to increased solubility and more efficient solvation processes.

In contrast to a sufficiently rapid change in the fluctuations of the mixture density with varying temperature and pressure, the difference in enthalpy occurs more smoothly, see Figure 6. The enthalpy has several local extrema; the first corresponds to the transition of ethanol to the gas phase, the second to the Widom delta of the mixture. Outside these zones, the dependence of enthalpy on temperature and pressure is monotonic. It is worth noting here a region with a rapid decrease in enthalpy in the vicinity of 400 K at a pressure of more than 60 bar, which also corresponds to a reduction in the enthalpy of mixing (from ~−4 J/mol at room temperature and atmospheric pressure to ~−8 J/mol; the free energy of mixing is on the order of −3 J/mol). This sharp decrease does not correspond to any Widom delta (neither the mixture nor the individual components). However, in this region, the best mixing of carbon dioxide with ethanol is achieved (see Figure 1c).

In contrast to enthalpy, its first derivative (specific isobaric heat capacity) has a much more complex behavior, reflecting all changes in the structure of matter: a transition to a supercritical state, a transformation from a single-phase to a two-phase system, a transition from a liquid-like to a gas-like SCF, etc. (see Figure 7).

The performed numerical simulation precedes the experimental study of the specific properties of the binary mixture under investigation using optical techniques that are simple, non-invasive, and sufficiently accurate. Such optical parameters of a medium as linear and nonlinear refractive indices depend on the nature of clustering of the medium [26]. Moreover, with an increase in the number of clusters of medium size with a pronounced axis, the molar refraction and the value of the nonlinear refractive index growth, as was previously shown on the example of pure carbon dioxide [26]. Thus, the observed appearance of large linear mixed clusters should lead to a sharp change in the medium’s optical properties. Change in the parameters of the medium also leads to a modification of the vibrational spectrum of the mixture. It can be verified, for example, by the method of Raman spectroscopy [48]. The growth of fluctuations in the Widom region can also be retrieved by measuring the laser beam intensity fluctuations transmitted through a medium [49]. Based on the results of the performed simulation, it is possible to schematically depict the observed Widom regions on the p-T diagram (see Figure 8). The growth of fluctuations increase the diffusion inside the media and can also enhance the rate of multiple chemical reactions or increase their efficiency [50]. However, the applied model does not describe the possible chemical reaction between the components, similarly to the chemical reaction between supercritical methanol and carbon dioxide on the inner surface of the cell [51].

The three observed Widom regions are expected to exist for several mixtures of linear molecules. The alignment of linear clusters is primarily due to a nonzero dipole moment in the ethanol molecule and a quadrupole moment in the carbon dioxide molecule. Thus, in the Widom region, they form extended linear clusters. In symmetric molecules (such as SF_6_) or noble gases (Ar, Ne, Xe), the effect may also appear but will be less pronounced. The formation of mixed CO_2_–C_2_H_5_OH clusters results from intermolecular interactions and is not limited to the investigated mixture. The complex mixtures can open a new route for new optical materials or efficient solvents. For instance, the nonlinear refractive index in the mixture Widom delta is triple of CO_2_ at the same pressure, due to the increase of form-factor γ (anisotropy factor) of linear clusters up to 0.8 (0.7 in pure CO_2_ Widom delta) [25]. It is also important to note that, formally, the Widom delta corresponding to the CO_2_ component lies below the critical point on the p-T diagram (see Figure 8). Such a behavior arises from formation of pure CO_2_ clusters. In this Widom delta, the mixture demonstrates properties similar to supercritical fluid such as high-density fluctuations and cluster formations. In a contrast, under temperatures above the critical point the maximal fluctuations and cluster formations only achieved in the mixture’s Widom delta. Therefore, the “formal” transition to the supercritical state does not guarantee the appearance of high fluctuations and cluster formation, and vice versa, the high fluctuations and cluster formations could be observed outside supercritical state.

In the development of the topic under discussion, alcohols such as ethanol or methanol are suitable solvents. Therefore, the use of deuterium-containing alcohols will allow organizing a mixture, including another substance (for example, liquid D_2_O) that can be initially dissolved in ethanol. Due to the effective diffusion of SCF carbon dioxide, the substance then spreads throughout the system, forming a monophase medium. This approach will allow us to perform experiments on the implementation of generating a neutron flux with the MeV energy level because of the action of super-intensive (more than 10^17^ W/cm^2^) femtosecond laser radiation on deuterium-containing aggregates [52]. This approach’s essence consists of the production of cluster jets during the supersonic expansion of supercritical matter into a vacuum for subsequent use as intense laser targets. Currently, the production of cluster jets during the supersonic expansion of a supercritical mater into a vacuum for subsequent use as laser targets and neutron generation is the subject of active research [40]. The developed methods allow us to analyze the thermodynamic state of such multicomponent mixtures.

## 3. Methods

One of the most common methods at an early stage of studying supercritical fluids’ properties is molecular dynamics (MD), which is a bridge between experimentally observed macroscopic properties of a substance with microscopic ones. This approach has been previously used to study the Widom delta separating the region of liquid-like from gas-like SCF in the p-T diagram [53,54,55]. Molecular dynamics (MD) is a method used for numerical simulation of the system evolution in time by integrating the equations of motion of atoms or particles. The movements of atoms or particles are calculated based on classical mechanics. Within the framework of the MD, the forces of interatomic interaction are represented in the form of the classical potential forces (the gradient of the potential energy of the system). The interatomic potential primarily determines the MD. The solution of the classical equations of motion in the framework of the MD leads to a set of trajectories that consist of atomic positions r and velocities v (or as an equivalent of momenta p) as a function of time *t* [56]. The thermodynamic parameters of the system are calculated as a function of these parameters. Since classical thermodynamics operates with continuous parameters formulated without considering the atomic (discrete) nature of matter, it does not connect thermodynamic quantities and atomic trajectories. Therefore, to retrieve TD parameters, it is necessary to use statistical thermodynamics (for equilibrium systems). The most used thermodynamic parameters are heat capacity at constant pressure (*C_p_*), isothermal compressibility (*β*), and coefficient of thermal expansion (*α*). In the numerical simulation, the heat capacity can be defined as:(1)$$C_{p}=\frac{H(p,T+\varepsilon )-H(p,T-\varepsilon )}{2\varepsilon },$$
where *H* is enthalpy, *p* is pressure, *T* is temperature, *V* is volume of system, and *ε* is small deviation (0.5 K and 0.1 bar in our numerical simulation). Isothermal compressibility and coefficient of thermal expansion are calculated similarly.
(2)$$\beta =-\frac{1}{V(p,T)}\frac{V(p+\varepsilon ,T)-V(p-\varepsilon ,T)}{2\varepsilon },$$
(3)$$\alpha =\frac{1}{V(p,T)}\frac{V(p,T+\varepsilon )-V(p,T-\varepsilon )}{2\varepsilon },$$

In this work, we used the LAMMPS software package [56]. The simulation was carried out for about 100,000 atoms with periodic boundary conditions, the interatomic interaction was specified by the COMPASS potential [57]. We used sixth power mixing rules and a long-range VanderWaals “correction” to the energy and pressure. The molecules were constructed with a MOLTEMPLATE utility [58]. The ethanol molecule was constructed based on the following script [59] and CO_2_ molecule from the following example [60]. The potential is an ab initio “class II” potential based on a matched force potential, which is successfully used to simulate both organic and inorganic molecules even at high (up to 2000 bar) pressures [61]. The choice of this potential was motivated by the best agreement between the calculated and experimental Raman spectra. The modeling was carried out in several stages. At the first stage, the system was brought into thermodynamic equilibrium (the criterion was the invariability of the internal energy and enthalpy of the system in 100,000 steps). To do this, we consistently simulated a microcanonical ensemble with a Langevin thermostat [62] and a Berendsen barostat [63] (fix NVE + fix Langevin + fix pressure/bredson) with a step of 1 femtosecond (fs). The combination of the applied fixes gives the best agreement with tabular values of density and enthalpy for pure CO_2_ and ethanol. For the initial achievement of TD equilibrium for a given temperature and atmospheric pressure, 100 million steps were taken. Further, for pressures with a step of 5 bar for a given temperature, the system was sequentially brought to the TD in equilibrium (fix NVE + fix Langevin + fix pressure/bredson) in 10 million steps (step is 0.1 fs). Then, for averaging over 100 points, the system’s density, volume, and enthalpy were calculated every 100,000 steps by built-in LAMMPS functions. Similar measurements were repeated for points on the p-T diagram at 0.1 bar and 0.5 K, respectively (4 points in total) for calculations by Equations (1)–(3). The Widom delta was determined as a region, where a maximum of fluctuations is reached [10], an extremum of the TD parameters is observed [17,18,19], and the maximum formation of clusters is achieved [32].

The applied numerical model was initially verified on pure CO_2_ and ethanol. The computed values of density and enthalpy were compared with tabular ones [64] (see Figure 9). Without vicinity of critical pressure, the calculated values are identical with tabular. In ethanol the calculated density and enthalpy for room conditions coincide with tabular data (0.785 kg/m^3^, 38.5 kJ/mol and 0.789 kg/m^3^, 38.56 kJ/mol respectively), the calculated critical point (507 K, 67 bar) is close to tabular (514 K, 63 bar). In critical point the difference between the calculated density/enthalpy with literature data (0.276 kg/m^3^ and 15.75 kJ/mol) [65] is higher (0.289 kg/m^3^ and 13.38 kJ/mol). The difference in the vicinity of the critical point is caused by slightly different values of critical parameters retrieved from molecular dynamics. Nevertheless, the observed differences do not change the effects of clustering qualitatively, only modify the Widom delta’s boundary.

## 4. Conclusions

We analyze the structure of a mixture containing carbon dioxide with ethanol (25% of the mixture) and the behavior of density, its fluctuations, enthalpy, and thermodynamic parameters in a wide range of temperatures and pressures including the supercritical state using molecular dynamics. It was revealed that there are three regions of significant (doubled compared with pure CO_2_) increase in density fluctuations, which are identified as Widom deltas. In these regions, the disintegration of the supercluster was observed, accompanied by a significant decrease in the number of molecules in it (more than three times), as well as an increase in the number of large linear mixed clusters (10–50 molecules). It has been shown that anomalous behavior of such parameters as enthalpy and isobaric heat capacity is a consequence of an increase in the number of medium-sized clusters and the decay of a supercluster corresponding to the liquid state of matter. It would be expected that the appearance of new Widom delta in the solvent mixture would lead to an increase in the reactivity of the complex solution. We believe that the complex mixtures can open a new route for the emergency of novel materials or efficient solvents. We expected that the additional information about microstructure of mixtures in such a promising region as Widom delta will find applications in various areas of science and technology. For example, increase in density fluctuations could increase the material removal rate during laser ablation; “the cluster formation in the Widom delta will increase the nonlinearity of the medium, which is interesting for generating a supercontinuum or for self-compression of ultrafast laser pulses, and the possible increase in the diffusivity of the mixture for the extraction of various organic and inorganic materials is also significant.

## Figures and Tables

**Figure 1 ijms-22-09813-f001:**
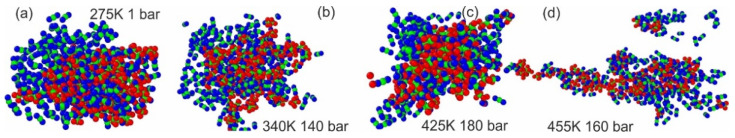
Visualization of the simulation results. Molecules of carbon, oxygen, and hydrogen are shown in green, blue, and red correspondingly. The figure shows the values of temperature and pressure. (**a**) Liquid ethanol with dissolved carbon dioxide with surrounding gaseous carbon dioxide; (**b**) the Widom delta of carbon dioxide; (**c**) the most compressed mixture with a uniform distribution of carbon dioxide and ethanol molecules; (**d**) the Widom delta of the mixture.

**Figure 2 ijms-22-09813-f002:**
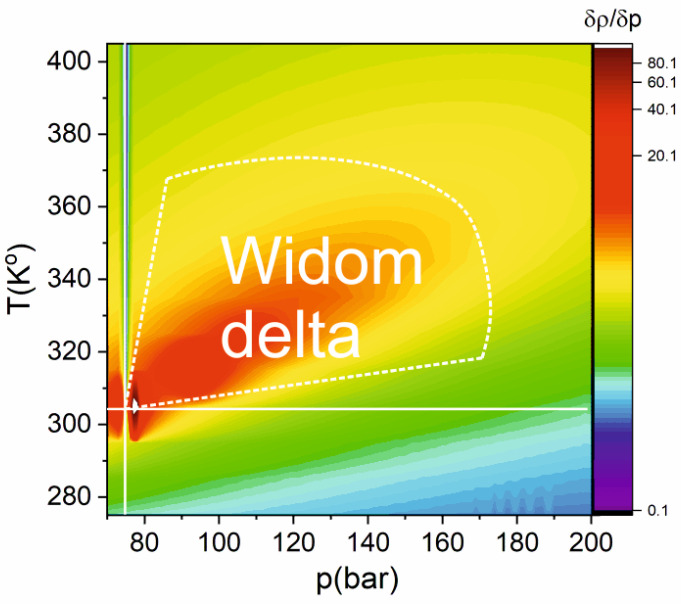
Heat map of the density derivative on pressure δρ/δp (logarithmic scale) in CO_2_. The dashed white line shows the Widom delta [43].

**Figure 3 ijms-22-09813-f003:**
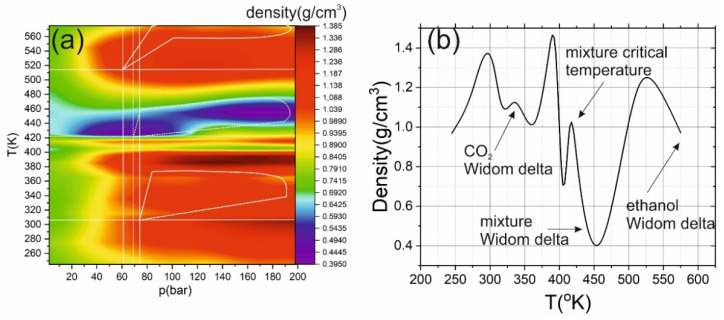
(**a**) Heat map of the mixture density dependence on pressure and temperature. The highlighted areas correspond to the Widom delta (solid lines for carbon dioxide and dashed lines for mixtures). (**b**) Dependence of density on temperature from Figure 1a under pressure of 161 bar. The solid lines show the critical temperatures and pressures of carbon dioxide and ethanol; the white dashed line shows the mixture critical point.

**Figure 4 ijms-22-09813-f004:**
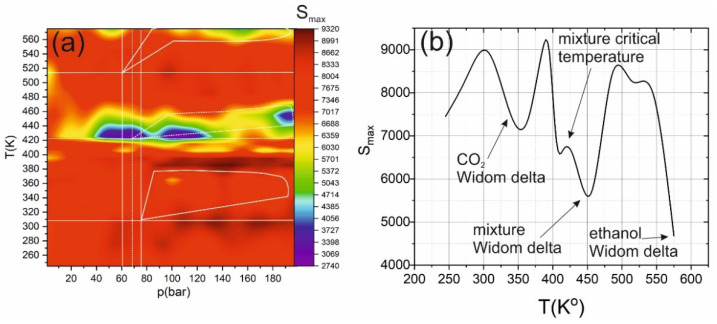
(**a**) Heat map of the maximum cluster size S_max_ (number of molecules) dependence on pressure and temperature. The solid lines show the critical temperatures and pressures of carbon dioxide and ethanol; the white dashed line shows the mixture critical point. The highlighted areas correspond to the Widom delta (solid lines for carbon dioxide and dashed lines for mixtures). (**b**) Dependence of the maximal cluster size on temperature under pressure of 161 bar.

**Figure 5 ijms-22-09813-f005:**
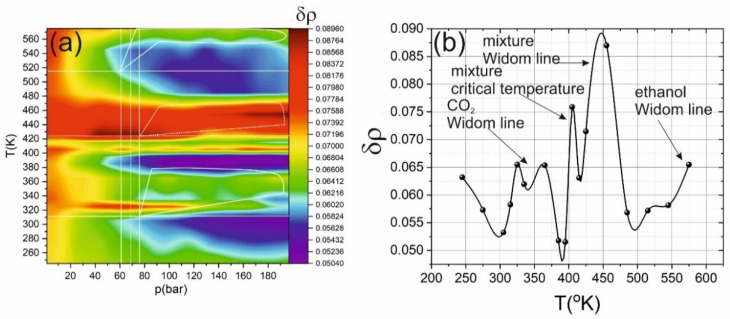
(**a**) Heat map of the density fluctuations δρ/δp dependence on pressure and temperature. Solid lines show the critical temperature and pressure of carbon dioxide and ethanol; dashed lines show mixture parameters. The highlighted areas correspond to the Widom delta (solid lines for carbon dioxide and dashed lines for mixtures). (**b**) The dependence of density fluctuations δρ on temperature from Figure 5a under pressure of 161 bar.

**Figure 6 ijms-22-09813-f006:**
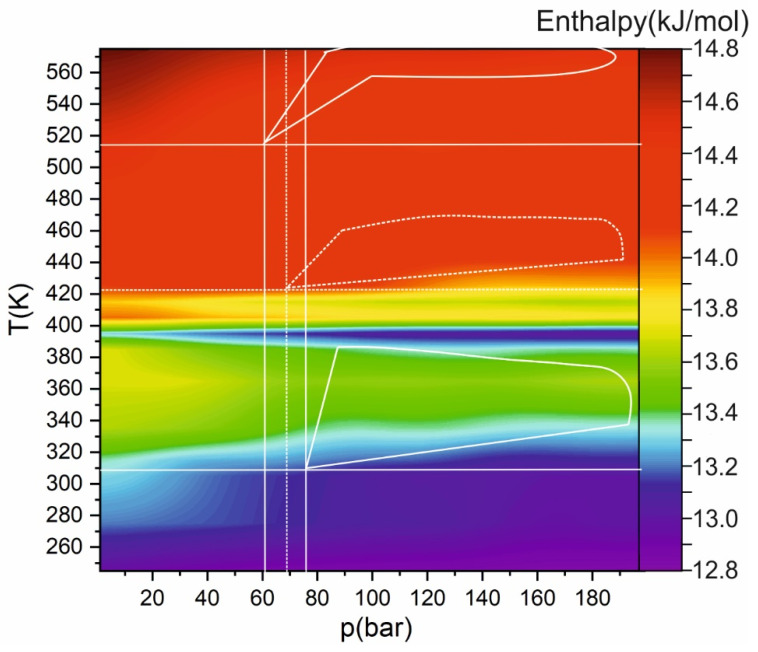
Heat map of enthalpy dependence on pressure and temperature. Solid lines show the critical temperature and pressure of carbon dioxide and ethanol; dashed lines correspond to critical mixture parameters. (Solid lines for carbon dioxide and dashed lines for mixtures.)

**Figure 7 ijms-22-09813-f007:**
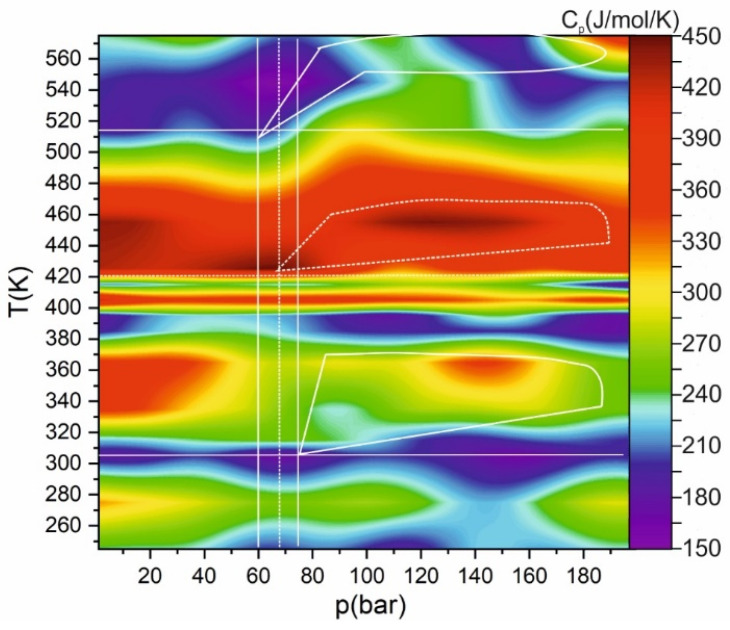
Heat map of specific isobaric heat capacity C_p_ on pressure and temperature. Solid lines show the critical temperature and pressure of carbon dioxide and ethanol; dashed lines show critical mixture parameters. The highlighted areas correspond to the Widom (solid lines for carbon dioxide and dashed lines for mixtures).

**Figure 8 ijms-22-09813-f008:**
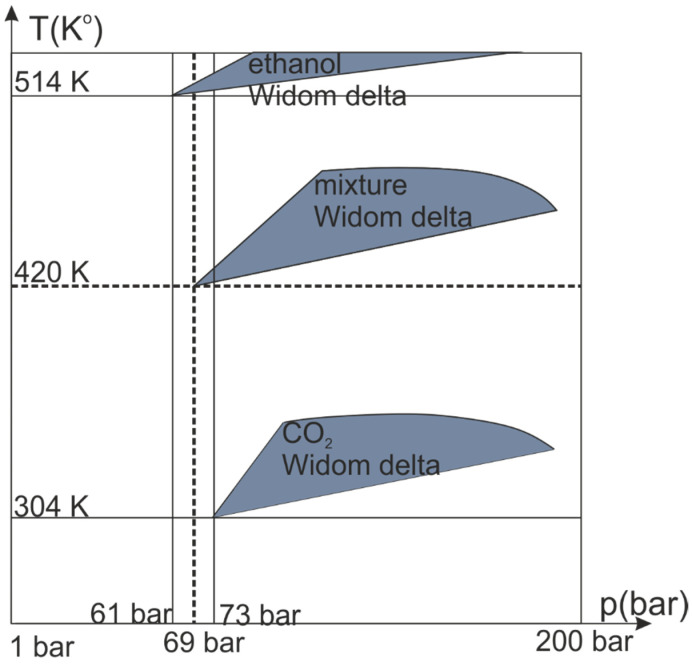
Schematic representation of a pressure–temperature diagram for a CO_2_–C_2_H_5_OH mixture. The dark regions show three Widom delta. Solid lines show the critical temperature and critical pressure for individual components. The dashed line shows critical temperature and critical pressure for the entire mixture.

**Figure 9 ijms-22-09813-f009:**
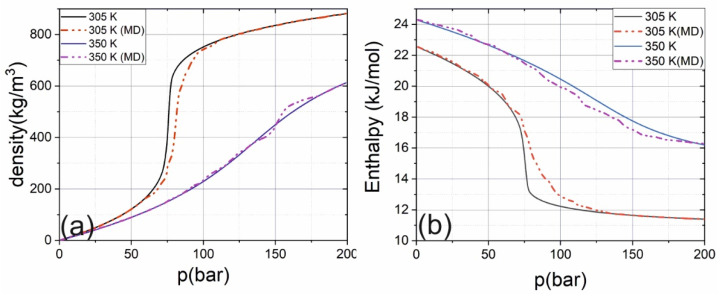
Dependence of computed (dash-dot lines) and tabular (solid lines) values of density (**a**) and enthalpy (**b**) on pressure for CO_2_ different temperatures.

## Data Availability

The data presented in this study are available on request from the corresponding author. The data are not publicly available due to privacy.

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
