# Peer review of "The Anomalous Behavior of Thermodynamic Parameters in the Three Widom Deltas of Carbon Dioxide-Ethanol Mixture"

_ijms, 2021, doi:10.3390/ijms22189813_

Round 1

Reviewer 1 Report

Comments

  1. Please clearly provide the computational details of present results.

  Please also define the equations.

(density, maximum cluster size, density fluctuations, enthalpy, specific isobaric heat capacity ……)

  1. The abbreviations were suggested to be defined at the first mention.

  Please use the abbreviations consistently.

  1. “……the 161-bar isobar……” (L142)

“……Smax = 9260……” (L161)

“……Solid lines indicate the critical temperature and pressure of carbon dioxide and ethanol, dashed lines – critical mixture parameters.……” (L230-231)

For readers’ comprehension, please also define clearly.

  1. Please provide the information of Fig 8 clearly.

  1. Please check the typing and grammar errors throughout the manuscript.

  1. In compare with previous reports (ex: Molecules 2020, 25:5424……), please highlight and improve the novelty of present study.

Please update the reference.

  1. Please check the format of citations.

(L39, 42……)

  1. Please improve the quality of manuscript, especially the abstract, methods and conclusions.

  1. Please use the same units throughout the manuscript.

  1. As we showed in [23], ……(L121)

   Please define clearly.

  1. ……the medium density and the maximum cluster size.” (L123-4)

   Please define clearly.

  1. “In addition, …expected that…In addition, …expected that…”

 (L346-350)

Author Response

First of all, our thanks to Reviewers for their valuable comments on our manuscript.

Below is the list of corrections we made in accordance with their suggestions. The full list of changes is presented in the redline version of the manuscript (MS).

Point 1:

Please clearly provide the computational details of present results. Please also define the equations. (density, maximum cluster size, density fluctuations, enthalpy, specific isobaric heat capacity ……)

Answer:

We added additional information for computational details: the used molecules templates, averaging information, and the method of computation. Thereby, in the current form of manuscript, the following information about performed simulations is presented: boundary conditions, potentials, number of atoms, molecule geometry, used fixes, computation time, etc.

We also defined all equations and figures.

 Point 2:

The abbreviations were suggested to be defined at the first mention. Please use the abbreviations consistently.

 Answer:

We defined abbreviations at the first mention.

Point 3:

“……the 161-bar isobar……” (L142)

“……Smax 9260……” (L161)

“……Solid lines indicate the critical temperature and pressure of carbon dioxide and ethanol, dashed lines – critical mixture parameters.……” (L230-231)

For readers’ comprehension, please also define clearly.

 Answer:

We fixed this text fragments.

“under pressure of 161 bar.”

“calculated cluster size (Smax) equals to 9260 molecules”

“The highlighted areas correspond to the Widom delta (solid lines for carbon dioxide and dashed lines for mixtures).”

 Point 4:

Please provide the information of Fig 8 clearly.

 Answer:

We fixed information of Fig. 8.

“Schematic representation of a pressure-temperature diagram for a CO2-C2H5OH mixture. The dark regions show three Widom delta. Solid lines show the critical temperature and critical pressure for individual components. The dashed line shows critical temperature and critical pressure for the entire mixture.”

 Point 5:

Please check the typing and grammar errors throughout the manuscript.

  Answer:

We tried to fix all errors in the manuscript.

 Point 6 and Point 8:

In compare with previous reports (ex: Molecules 2020, 25:5424……), please highlight and improve the novelty of present study. Please update the reference. 

Please improve the quality of manuscript, especially the abstract, methods and conclusions.

 Answer:

The work [Molecules 2020] contains information about increase of nonlinear properties of CO2-ethanol mixture in the Widom delta of this mixture. However, the behavior of thermodynamic parameters of this mixture have never been considered before. According to the fact that the first mentioning of the third Widom delta is given in previous our article we changed the tittle to “The anomalous behavior of thermodynamic parameters in the three Widom deltas of supercritical carbon dioxide-ethanol mixture”.

But we do not agree with the opinion about insufficient novelty of the study:

  1. We revealed that there is a new region that corresponds to the third Widom delta of mixture with preserving of Widom deltas of individual components
  2. We demonstrated the anomalous behavior of thermodynamic parameters in these regions
  3. We showed that origin of Widom delta and anomalous behavior of TD parameters it is a result of formation of large mixed clusters

In our opinion that is completely new results reveling connection between macroscopic and microscopic parameters of the mixture.

 Nevertheless, we rewrote Abstract, Introduction and conclusion. We also added additional information to the Methods (see Point 1).

 Point 7:

Please check the format of citations. (L39, 42……)

 Answer:

We checked and fixed the citations.

  Point 9:

Please use the same units throughout the manuscript.

  Answer:

In the current version of the manuscript we used the same units (K, bar, J/mol)

  Point 10:

As we showed in [23], ……” (L121)    Please define clearly.

   Answer:

There was a typo in the reference (it was non- Mendeley format). We fixed the reference.

  Point 11:

……the medium density and the maximum cluster size.” (L123-4)   Please define clearly.

  Answer:

We rewrote the sentence: “. Therefore, at the first stage, the MD method was used to study the dependence of the medium density (r) and the maximum clusters size (Smax) on pressure and temperature”

  Point 12:

“In addition, …expected that…In addition, …expected that…”

Answer:

We fixed the typo.

Reviewer 2 Report

This paper contains new results of a wider research on anomalous behavior of supercritical fluids in Widom regions (see refs. 28, 29, 52, and 58). Molecular dynamics was applied to simulate a mixture of carbon dioxide and ethanol in the temperature range wider than 250-570 K and pressure range up to more than 190 bar. Thermodynamic properties were determined from steady-state results of simulations. Three Widom regions were located and characterized. The results are clearly shown in three-dimensional maps.

The results are certainly worth of publishing; the paper itself needs minor corrections. Authors should check whether all references in the text, given by numbers, agree with the list References. There seem to be a few discrepancies. Namely, ref. 23 on line 121 and probably also ref. 30 on line 303 are not correct. “see Fig. 9” should be written on line 326. The position of the left-hand edge of CO2 Widom region in Fig. 8 should be shifted.

I am not sure whether I understand well the sentences on lines 62-66: “Crucially, that the addition of even a tiny amount of another substance can significantly shift the critical parameters of this system (up to hundreds of degrees and bars), in a first approximation, the critical temperature and pressure of the resulting mixture can be estimated as partial values of the critical parameters of each component.“  A tiny amount of another substance shifts the critical point by a large extent, and still the new critical point can be estimated as partial values of critical parameters of the major component and tiny impurity?  

Further suggested corrections

Line 78: In particular they used in the process - In particular they were used in the process

Line 102: Fig.1a. also shows - Fig. 1a also shows

Lines 135-138: Please reformulate these sentences: “The further the experimental conditions on the p-T diagram are from the critical point, the wider the peak becomes in the dependence in Fig. 2. The lower its amplitude, it is most pronounced for a pressure of 79 bar and a temperature of 308 K.“

Line 173: as efficiently as pure  - as efficiently as in pure

Line 174: The microstructure of – In the microstructure of

Lines 222-227: Decrease or increase? “It is worth noting here a region with a rapid decrease in enthalpy in the vicinity of 400 K at a pressure of more than 60 bar, which also corresponds to a reduction in the enthalpy of mixing -4 J/mol at room temperature and atmospheric pressure to ~ -8 J/mol, the free energy of mixing is on the order of -3 J/mol). This sharp increase does not correspond to any Widom delta (neither the mixture nor the individual components).”

Lines 258/9: (A verb is missing in the sentence.)

Line 271: such as ethyl or methyl - such as ethanol or methanol

Line 330: only modifies - only modify

Lines 334/5: doubled compare with - doubled compared with

Line 405: Wiodm - Widom

Line 421: CO<inf>2</inf>and – CO2 and

Line 425: Zav’Yalov - Zav’yalov

Author Response

First of all, our thanks to Reviewers for their valuable comments on our manuscript.

Below is the list of corrections we made in accordance with their suggestions. The full list of changes is presented in the redline version of the manuscript (MS).

Point 1:

 There seem to be a few discrepancies. Namely, ref. 23 on line 121 and probably also ref. 30 on line 303 are not correct. “see Fig. 9” should be written on line 326. The position of the left-hand edge of CO2 Widom region in Fig. 8 should be shifted.

  Answer:

There was a typo in the references (they were non- Mendeley format). We fixed them. The numeration of Figures was corrected. We redraw Fig.8.

Point 2:

I am not sure whether I understand well the sentences on lines 62-66: “Crucially, that the addition of even a tiny amount of another substance can significantly shift the critical parameters of this system (up to hundreds of degrees and bars), in a first approximation, the critical temperature and pressure of the resulting mixture can be estimated as partial values of the critical parameters of each component.“  A tiny amount of another substance shifts the critical point by a large extent, and still the new critical point can be estimated as partial values of critical parameters of the major component and tiny impurity?  

Answer:

If there is a huge difference between the critical parameters of the two components, the mixture critical point (proportional as partial value of critical parameters) could significantly be changed. However we reformulate the sentence: “Crucially, that the addition of even a tiny amount of another substance with different critical pressure and temperature can significantly shift the critical parameters of this system (up to hundreds of degrees and bars), because in a first approximation, the critical temperature and pressure of the resulting mixture can be estimated as partial values of the critical parameters of each component”

Point 3,4:

Line 78: In particular they used in the process - In particular they were used in the process

Line 102: Fig.1a. also shows - Fig. 1a also shows

Answer:

We fixed these text fragments.

Point 5:

Lines 135-138: Please reformulate these sentences: “The further the experimental conditions on the p-T diagram are from the critical point, the wider the peak becomes in the dependence in Fig. 2. The lower its amplitude, it is most pronounced for a pressure of 79 bar and a temperature of 308 K.“

Answer:

We rewrote the sentence: “At a longer distance the experimental conditions on the p-T (pressure-temperature) diagram from the critical point, the peak of density derivative on pressure becomes wider (Fig. 2.) and its amplitude is decreased. The maximal amplitude is achieved for a pressure of 79 bar and a temperature of 308 K.”

Point 6,7:

Line 173: as efficiently as pure  - as efficiently as in pure

Line 174: The microstructure of – In the microstructure of

Answer:

We fixed these text fragments.

Point 8:

Lines 222-227: Decrease or increase? “It is worth noting here a region with a rapid decrease in enthalpy in the vicinity of 400 K at a pressure of more than 60 bar, which also corresponds to a reduction in the enthalpy of mixing -4 J/mol at room temperature and atmospheric pressure to ~ -8 J/mol, the free energy of mixing is on the order of -3 J/mol). This sharp increase does not correspond to any Widom delta (neither the mixture nor the individual components).”

Answer:

Increase. We fixed the text fragment.

Point 9, 10, 11, 12, 13, 14, 15:

Lines 258/9: (A verb is missing in the sentence.)

Line 271: such as ethyl or methyl - such as ethanol or methanol

Line 330: only modifies - only modify

Lines 334/5: doubled compare with - doubled compared with

Line 405: Wiodm - Widom

Line 421: CO<inf>2</inf>and – CO2 and

Line 425: Zav’Yalov - Zav’yalov

Answer:

We fixed these typos.

Reviewer 3 Report

The manuscript by Mareevv, Sviridov, and Gordienko performs MD simulations for supercritical CO2-ethanol mixture. They propose that there exists three Widom deltas where the thermodynamic response functions (or density fluctuations) have their maxima. The article itself is very interesting, and worthwhile for the publication. I would like to ask the following questions.

(1) In the abstract (line 14), I guess "interrupted" should be changed to "interpreted".

(2) Although the authors describe the MD simulation procedure they used, I think more detailed description is required so that the manuscript itself can answer the following questions.

(a) Why did you choose COMPASS potential? For CO2, two models, including the EPM2 model and the TraPPE model, are normally adopted. What about ethanol?

(b) In line 303, "CO2" should be changed to CO2.

(c) Why did you use langevin thermostat, not other thermostats? Of course, I guess the thermostat selection would not have a significant impact on the qualitative picture, but langevin thermostat is known to yield inaccurate dynamic properties, since it imposes random motion to molecules. (I guess its impact on the thermodynamic functions should be small, though)

(d) What mixing rule was used? Lorentz-Berthelot? geometric mixing rule? or others?

(e) The authors included the pure CO2 results. What about ethanol? What would be the critical point of pure CO2 and pure ethanol model adopted?

(3) In line 170-172, the authors say they defined the local density by volume. What types of algorithms did they use? If it is a spatial tessellation technique like Voronoi tessellation, the authors should clarify what types of tessellation technique they used. For instance, simple Voronoi tessellation and Radical Voronoi tessellation results are slightly different from each other. Also, if the authors chose either of them, they should justify their selection. For instance, the authors can see the difference between different spatial tessellation techniques in the article by Yoon (The Journal of chemical physics 150 (15), 154503)

(4) The authors show that there are three distinct regions. What would be the relation among them? Can we "infer" the mixture Widom delta from CO2 Widom delta and ethanol Widom delta?

Author Response

First of all, our thanks to Reviewers for their valuable comments on our manuscript.

Below is the list of corrections we made in accordance with their suggestions.

Point 1:

In the abstract (line 14), I guess "interrupted" should be changed to "interpreted".

Answer:

We fixed the typo.

 Point 2:

Why did you choose COMPASS potential? For CO2, two models, including the EPM2 model and the TraPPE model, are normally adopted. What about ethanol?

 Answer:

There are two main reasons for choosing the COMPASS potential. They are same for CO2 and ethanol.

  1. With the reax-ff potential it gives the best results for simulation of the Raman spectrum and gives a good agreement with tabular data (density behavior, enthalpy, critical point). And it requires fewer computing resources than reax-ff.
  2. It gives the best results for CO2-ethanol interaction.

Following the comment we added the information about our choice motivation to the text.

Point 3:

In line 303, "CO2" should be changed to CO2.

Answer:

We fixed the typo.

Point 4:

Why did you use langevin thermostat, not other thermostats? Of course, I guess the thermostat selection would not have a significant impact on the qualitative picture, but langevin thermostat is known to yield inaccurate dynamic properties, since it imposes random motion to molecules. (I guess its impact on the thermodynamic functions should be small, though)

 Answer:

The reason is the same as for the Point 2. The applied combination of thermostats and barostats gives the best result for modeling of Raman spectrum and critical density of pure CO2 and ethanol. We have tried several combinations and the applied one gave the best result.

We added the motivation of our choice to the text:
“The combination of the applied fixes gives the best agreement with tabular values of density and enthalpy for pure CO2 and ethanol.”

 Point 5:

What mixing rule was used? Lorentz-Berthelot? geometric mixing rule? or others?

 Answer:

We used sixthpower mixing rules. We added the information to the text.

Point 6:

The authors included the pure CO2 results. What about ethanol? What would be the critical point of pure CO2 and pure ethanol model adopted?

 Answer:

Unfortunately, there is no information about fluid properties of ethanol for different range of pressures and temperatures in https://webbook.nist.gov/chemistry/, thereby we do not find tabular data for ethanol density and enthalpy for investigated isotherms. But following the recommendations for verifying of our model we presented data for room temperature and critical point of ethanol. In ethanol the calculated density and enthalpy for room conditions coincide with tabular data (0.785 kg/m3, 38.5kJ/mol and 0.789 kg/m3 and 38.56kJ/mol respectively), the calculated critical point (507K, 67bar) is close to tabular (514 K, 63 bar). In critical point the difference between the calculated density/enthalpy with literature data (0.276kg/m3 and 15.75 kJ/mol) [Marcus, Y. Extraction by subcritical and supercritical water, methanol, ethanol and their mixtures. Separations 2018, 5, doi:10.3390/separations5010004.] is higher (0.289kg/m3 and 13.38 kJ/mol). We added the presented above fragment to the text of the MS. The calculated critical point of CO2 is 301 K and 71 bar.

 Point 7:

In line 170-172, the authors say they defined the local density by volume. What types of algorithms did they use? If it is a spatial tessellation technique like Voronoi tessellation, the authors should clarify what types of tessellation technique they used. For instance, simple Voronoi tessellation and Radical Voronoi tessellation results are slightly different from each other. Also, if the authors chose either of them, they should justify their selection. For instance, the authors can see the difference between different spatial tessellation techniques in the article by Yoon (The Journal of chemical physics 150 (15), 154503)

 Answer:

We used the Radical Voronoi tessellation using the VoroTop ++ package. Following the suggested article [(The Journal of Chemical physics 150 (15), 154503)] it seems that it is a better choice for Widom deltas.

  Point 8:

The authors show that there are three distinct regions. What would be the relation among them? Can we "infer" the mixture Widom delta from CO2 Widom delta and ethanol Widom delta?

 Answer:

Unfortunately, in our opinion there is no direct connection between the location of the third Widom delta and Widom deltas of pure components. It is specific for each molar fraction and the mixture`s composition. Without a doubt, one can only assert that the third Widom delta would be between the “pure” deltas. Moreover, for components with similar critical points it will be closer to each one. We will try to find the relation between the deltas in our future work.

Reviewer 4 Report

Manuscript ijms-1312844 has been very well written. Despite the complexity of the subject matter, the authors try to clarify new concepts, which is very helpful to the reader. In addition, I would like to highlight the relevance of the references used. My main concern lies in the novelty of the work, which could have been compromised with a previous publication by the authors themselves: Molecules 2020, 25(22), 5424; https://doi.org/10.3390/molecules25225424. I say the above, considering that the authors use the same study methodology, with the same mixture. We are talking about a publication in a very high-impact journal such as IJMS, so the novelty of the work cannot be overlooked. However, that is a consideration that I leave in the hands of the Editor.

On the other hand, as the same authors indicate at the beginning of the R & D section, a mixture CO2 + 25 % EtOH is not supercritical. At these proportions, we would be talking about an expanded liquid. This is a major consideration that must be considered when giving the title to the work and describing it.

Finally, I would like to see a more in-depth methodological description, which would help the reproducibility of the work. The authors could be helped by citing other work for this. In the case of conclusions, the authors could provide more future perspectives.

Other considerations:

L57: Indicate other applications, if any. This will help to justify the work.

L87: Cite works.

- The authors leave the first title of Results and discussions, followed by the second title of discussions. I would prefer that there be only one for R&D.

- I recommend authors enter Figures after citing them. The current manuscript does not fulfill this.

Author Response

First of all, our thanks to Reviewers for their valuable comments on our manuscript.

Below is the list of corrections we made in accordance with their suggestions. The full list of changes is presented in the redline version of the manuscript (MS).

Point 1:

My main concern lies in the novelty of the work, which could have been compromised with a previous publication by the authors themselves: Molecules 2020, 25(22), 5424; https://doi.org/10.3390/molecules25225424. I say the above, considering that the authors use the same study methodology, with the same mixture. We are talking about a publication in a very high-impact journal such as IJMS, so the novelty of the work cannot be overlooked. However, that is a consideration that I leave in the hands of the Editor.

Answer:

 The work [Molecules 2020] contains information about increase of nonlinear properties of CO2-ethanol mixture in the Widom delta of this mixture. However, the behavior of thermodynamic parameters of this mixture have never been considered before.

We want to additionally clarify the novelty of the study:

  1. We revealed that there is a new region that corresponds to the third Widom delta of mixture with preserving of Widom deltas of individual components
  2. We demonstrated the anomalous behavior of thermodynamic parameters in these regions
  3. We showed that origin of Widom delta and anomalous behavior of TD parameters it is a result of formation of large mixed clusters

In our opinion, that is completely new results reveling connection between macroscopic and microscopic parameters of the mixture.

 Following the comment, we rewrote Abstract, Introduction and conclusion

Point 2:

On the other hand, as the same authors indicate at the beginning of the R & D section, a mixture CO2 + 25 % EtOH is not supercritical. At these proportions, we would be talking about an expanded liquid. This is a major consideration that must be considered when giving the title to the work and describing it.

Answer:

We changed the tittle to “The anomalous behavior of thermodynamic parameters in the three Widom deltas of supercritical carbon dioxide-ethanol mixture”.  We believe that in its current form, the use of the term supercritical for a mixture is justified due to the fact that the Widom delta is a specific for supercritical fluids.

Point 3:

Finally, I would like to see a more in-depth methodological description, which would help the reproducibility of the work. The authors could be helped by citing other work for this.

Answer:

We added additional information for computational details: the used molecules templates, averaging information, and the method of computation. Thereby, in the current form of manuscript, the following information about performed simulations is presented: boundary conditions, potentials, number of atoms, molecule geometry, used fixes, computation time, etc. “

Point 4:

 In the case of conclusions, the authors could provide more future perspectives.

Answer:

We added the possible perspective applications of the fluids with pressure and density corresponding to the Widom delta. We added the following fragment to the conclusion:
“It would be expected that the appearance of new Widom delta in the solvent mixture would lead to an increase in the reactivity of the complex solution. We believe that the complex mixtures can open a new route for the emergency of novel materials or efficient solvents. We expected that the additional information about microstructure of mixtures in such a promising region as Widom delta will find applications in various areas of science and technology. For example, increase in density fluctuations could increase the material removal rate during laser ablation; "the cluster formation in the Widom delta will increase the nonlinearity of the medium, which is interesting for generating a supercontinuum or for self-compression of ultrafast laser pulses, and the possible increase in the diffusivity of the mixture for the extraction of various organic and inorganic materials is also significant.”

Point 5:

L57: Indicate other applications, if any. This will help to justify the work.

Answer:

We added the reference to the review [K. Byrappa, S. Ohara, and T. Adschiri, "Nanoparticles synthesis using supercritical fluid technology - towards biomedical applications," Adv. Drug Deliv. Rev. 60(3), 299–327 (2008).] about nanoparticles productions in supercritical fluids. However, in our knowledge there is no information about nanoparticles production in Widom delta.

 Point 6:

L87: Cite works.

Answer:

We rewrote the introduction and there is no such sentence in this version of the manuscript, however we add additional reference to line 74.

Point 7:

- The authors leave the first title of Results and discussions, followed by the second title of discussions. I would prefer that there be only one for R&D.

Answer:

We jointed the Results and Discussion sections

 Point 8:

- I recommend authors enter Figures after citing them. The current manuscript does not fulfill this.

Answer:

We changed the locations of the Figures in the text of the MS.

Round 2

Reviewer 4 Report

A CO2 + 25 % ethanol mixture is not supercritical and the authors must address this issue.

Author Response

Point 1:

A CO2 + 25 % ethanol mixture is not supercritical and the authors must address this issue.

  Answer:

Thank you for this non-trivial issue. It helps us to interpret some data in alternative way. Indeed, the mixture is not supercritical in the most part of the p-T diagram, even above the mixture`s critical point. However, it is generally considered that each material with pressure and temperature above the critical point must be treated as supercritical fluid. This is also true for mixtures, see several papers [J. W. Ziegler, T. L. Chester, D. P. Innis, S. H. Page, and J. G. Dorsey, "Supercritical Fluid Flow Injection Method for Mapping Liquid—Vapor Critical Loci of Binary Mixtures Containing CO2," in In Innovations in Supercritical Fluids (1995), pp. 93–110.;           H. Pöhler and E. Kiran, "Volumetric properties of carbon dioxide + ethanol at high pressures," J. Chem. Eng. Data 42(2), 384–388 (1997).; A. M. Palma, A. J. Queimada, and J. A. P. Coutinho, "Modeling of the Mixture Critical Locus with a Modified Cubic Plus Association Equation of State: Water, Alkanols, Amines, and Alkanes," Ind. Eng. Chem. Res. 57(31), 10649–10662 (2018).; S. Do Yeo, S. J. Park, J. W. Kim, and J. C. Kim, "Critical properties of carbon dioxide+methanol, +ethanol, +1-propanol, and +1-butanol," J. Chem. Eng. Data 45(5), 932–935 (2000).; J. Ke, P. J. King, M. W. George, and M. Poliakoff, "Method for locating the vapor-liquid critical point of multicomponent fluid mixtures using a shear mode piezoelectric sensor," Anal. Chem. 77(1), 85–92 (2005).]. Nevertheless the “supercritical” properties (cluster formation, high fluctuations, etc.) are only obtained in the Widom deltas, and one of the Widom deltas is below the critical point (for pure CO2). Based on this fact we excluded the term supercritical from the tittle and abstract. We also added to the discussion the following text fragment:
“In this Widom delta the mixture demonstrates properties similar to supercritical fluid such as high density fluctuations and cluster formations.  In a contrast, under temperatures above the critical point the maximal fluctuations and cluster formations only achieved in the mixture`s Widom delta. Thereby the “formal” transition to the supercritical state does not guarantee the appearance of high fluctuations and cluster formation, and vice versa, the high fluctuations and cluster formations could be observed outside supercritical state.”

Round 3

Reviewer 4 Report

I am glad the authors have reconsidered their position.